# LLMs Pick Up Cues of Potential Comorbid ADHD in People Reporting Anxiety when Keywords Are Not Enough[*]

**Claire S. Lee[1], Noelle Lim[2, 3], and Michael Guerzhoy[3]**

[1]Princeton University [2]LinkedIn Corporation [3]University of Toronto

## Abstract

We present a novel task that can elucidate the connection between anxiety and ADHD; use Transformers to make progress toward solving a task that is not solvable by keyword-based classifiers; and discuss a method for visualization of our classifier illuminating the connection between anxiety and ADHD presentations.

Up to approximately 50% of adults with ADHD may also have an anxiety disorder and approximately 30% of adults with anxiety may also have ADHD. Patients presenting with anxiety may be treated for anxiety without ADHD ever being considered, possibly affecting treatment. We show how data that bears on ADHD that is comorbid with anxiety can be obtained from social media data, and show that Transformers can be used to detect a proxy for possible comorbid ADHD in people with anxiety symptoms.

We collected data from anxiety and ADHD online forums (subreddits). We identified posters who first started posting in the Anxiety subreddit and later started posting in the ADHD subreddit as well. We use this subset of the posters as a proxy for people who presented with anxiety symptoms and then became aware that they might have ADHD. We fine-tune a Transformer architecture-based classifier to classify people who started posting in the Anxiety subreddit and then started posting in the ADHD subreddit vs. people who posted in the Anxiety subreddit without later posting in the ADHD subreddit. We show that a Transformer architecture is capable of achieving reasonable results (76% correct for RoBERTa vs. under 60% correct for the best keyword-based model, both with 50% base rate).

## Introduction

Up to 53% of adults with ADHD may also have an anxiety disorder (Children and with Attention-Deficit/Hyperactivity Disorder Report 2018) (Quinn and Madhoo 2014) and up to 28% of adults who have an anxiety disorder may also have ADHD (Van Ameringen et al. 2011). However, patients presenting with anxiety or depressive symptoms may be treated for these disorders without ADHD ever being considered

(Quinn and Madhoo 2014) (Katzman et al. 2017). Misdiagnosis of ADHD is common, as many clinicians are still not aware that ADHD is a valid diagnosis in adults (Quinn and Madhoo 2014) and physicians are more familiar with mood and anxiety disorders (Katzman et al. 2017). The danger of misdiagnosed comorbid ADHD and anxiety is that only the symptoms for anxiety will be treated and ADHD will be left untreated (Katzman et al. 2017). Social media such as Reddit provides publicly available text data of anonymous first-person experiences (Low et al. 2020).

We analyze people talking about their mental health on the forum website Reddit. We propose classifying posts from people who only posted in the Anxiety subreddit (forum) and never in the ADHD subreddit vs people who posted in the Anxiety subreddit and will later have started posting in the ADHD subreddit. This way, we can distinguish text from users whose posting will show interest/concern with ADHD in the future from people whose posting will not do that. Posting about ADHD is a proxy for being concerned about ADHD. Showing that this task is possible indicates that there is a systematic difference between the two groups of Reddit users. Our hope is that analyzing the classifier can elucidate the connection between anxiety and anxiety-comorbid ADHD. A limitation is that posting on Reddit is a proxy for concern with ADHD, and there can be both false positives and false negatives if Reddit posting is used as a proxy for identifying patients with ADHD.

We demonstrate that the task above is not solvable using keyword-based methods such as Naive Bayes and logistic regression. We then demonstrate that the task is better solved using RoBERTa (Liu et al. 2019), indicating that the connection between anxiety and anxiety-comorbid ADHD is more complex than what can be captured with keywords. We report on visualizing the "explanation" for the classifier's prediction. In future work, we plan to use this visualization to gain insight into the connection between anxiety and anxiety-comorbid ADHD.

Transformer models such as RoBERTa have been used to classify mental health disorders from social media text (Ameer et al. 2022) (Murarka, Radhakrishnan, and Ravichandran 2020).

In the rest of the paper, we discuss our data collection process and report on our experiments showing that it is possible to predict which posts in the Anxiety subreddit come

---

[*]A version of this paper was published at CLPsych @ EACL under the title "Detecting a Proxy for Potential Comorbid ADHD in People Reporting Anxiety Symptoms from Social Media Data".

from people who will never post in the ADHD subreddit vs. people who will post in the ADHD subreddit. In the Appendix, we expand on comorbid ADHD and anxiety, and using learned classifiers on mental health-related text.

## Data Collection

Text data was collected from the Anxiety and ADHD subreddits on Reddit. Although Reddit posts are not formal clinical diagnoses, Reddit data offers advantages such as being immediately and publicly available, including a timeframe to track historical data, and anonymous posts documenting vulnerable first-person experiences (Low et al. 2020).

All posts were scraped from the Anxiety and ADHD subreddits from the dates February 16, 2020 to Nov 28, 2022.

### Data Preprocessing

The data was cleaned by removing empty or removed posts. The data was filtered to only contain posts from users that only ever posted in the Anxiety subreddit or who first posted on the Anxiety subreddit only then in the ADHD subreddit.

For users who started posting in the ADHD subreddit eventually, we only kept posts from the Anxiety subreddit that were posted 6 months or more before the first post in the ADHD subreddit. No posts from the ADHD subreddit were used.

In total, 47482 posts were downloaded from the ADHD and Anxiety subreddits. 33% were retained for the test set.

## Models

### Baseline Model

Our baseline models were regularized logistic regression and binomial Naive Bayes.

**Transformer Model** We fine-tined the pre-trained RoBERTa model from HuggingFace (Huggingface 2020) with the RoBERTa tokenizer, the cross-entropy loss function, the Adam optimizer with a learning rate of $1e-5$, and a dropout layer with $p = 0.3$.

## Results

### Baseline Results

With a baserate of 50%, the best logistic regression model achieved a correct classification rate of 54% and the best Naive Bayes model achieved a correct classification rate of 59%.

As seen, Logistic Regression models performed at 54% accuracy and Naive Bayes performed at $58.6\%$ accuracy. Attributing to its performance, as seen in Figure 6, the majority of samples fell correctly into the true positive and true negative class.

### RoBERTa Results

With a test set baserate of 50%, the RoBERTa model achieved a correct classification rate of 76%.

## Discussion

Our results demonstrate that, for posts in the Anxiety subreddit, it is possible to predict which posts come from people who will later post in the ADHD subreddit as well from posts that come from people who will not, without using any information "from the future."

Further, we have shown that keyword-based methods, Naive Bayes and logistic regression, are not sufficient for this task, while it is possible to make progress with RoBERTa. This indicates that complex cues can be used to detect which posters will later post in the ADHD subreddit.

## Experiments with explainability

One application of our trained classifier is obtaining further insight into the relationship between anxiety disorders and ADHD.

To enable qualitative analysis, we have experimented with visualizing the reason that the RoBERTa classifier outputs "will post in ADHD" or "will not post in ADHD" for a given post. We visualize the difference in output caused by masking out each individual word and each individual phrase in the post.

Aggregate analysis will be available in the future.

## Conclusions

We present a novel task: predicting whether internet text that comes from a person discussing their anxiety comes from a person who in the future will also discuss ADHD. We demonstrate that this task is not solvable using keyword-based methods, while it progress can be made using RoBERTa.

The immediate application of our method is for obtaining qualitative insight into the connection between anxiety and ADHD by visualizing the reason that the RoBERTa classifier outputs "will post in ADHD" or "will not post in ADHD" for a given post.

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

# Background

## Comorbid ADHD and Anxiety

Up to 53% of adults with ADHD may also have an anxiety disorder (Children and with Attention-Deficit/Hyperactivity Disorder Report 2018) and about 3 in 10 children with ADHD had anxiety [1]. ADHD and anxiety are frequently seen together; however, ADHD and anxiety are separate conditions (Ellis 2017). ADHD is a common mental health disorder with symptoms such as inattention (not being able to keep focus), hyperactivity, and impulsivity (Elmaghraby and Garayalde 2022). Anxiety disorders may involve symptoms of excessive fear (Muskin 2022) which does not go away over time (NIMH 2023). People who have anxiety disorders may struggle with intense and uncontrollable feelings of anxiety, fear, worry, or panic (American Psychiatric Association, Association et al. 2013).

Although ADHD and anxiety have different symptoms, there are instances when the two conditions have overlapping symptoms, making it difficult to distinguish between the two conditions (Story 2022). For instance, individuals with anxiety may have trouble concentrating in situations that trigger anxiety. Those with ADHD may have trouble concentrating in any type of situation (Story 2022).

It is important to correctly diagnose patients with ADHD and anxiety, as the co-occurrence of ADHD and anxiety may make the symptoms of both conditions seem more extreme (Koyuncu et al. 2022). For example, anxiety can make it even more difficult for someone with ADHD to pay attention and follow through on tasks (Story 2022).

Misdiagnosed comorbid ADHD and anxiety may lead to treating only the symptoms for anxiety while the root of the problem, which is ADHD, remains untreated (Katzman et al. 2017) (Hallowell 2018). Undiagnosed ADHD can cause anxiety and depression which, in turn, can mask ADHD, making it more difficult to diagnose accurately (Kistler 2022).

Diagnosis of ADHD typically occurs in children; however, ADHD is now recognized to be persistent to adulthood in 50-66% of people (Johnson, Morris, and George 2020). Misdiagnosis of ADHD is common as many clinicians are still not aware that ADHD is a valid diagnosis in adults (Johnson, Morris, and George 2020).

## Classification of Mental Health-related texts with Deep Learning

Machine learning techniques have been utilized for multi-class classification of mental health condition-related text, particularly on Reddit. (Ameer et al. 2022) trained various models to detect texts related to anxiety, ADHD, bipolar disorder, depression, and PTSD. Ameer et al. observed that a pre-trained and then fine-tuned RoBERTa classifiers achieved the best performance. (Murarka, Radhakrishnan, and Ravichandran 2020) used RoBERTa to detect and classify texts related to mental health conditions and observed better accuracy and F-1 scores than BERT or LSTM models.

---

[1]https://www.cdc.gov/ncbddd/adhd/data.html

However, their model performed poorly in classifying anxiety, partially due to the term "anxiety" occurring in 12% of ADHD posts.

The dataset that was used for both papers was data scraped from 13 subreddits using the Reddit API: 17,159 posts and title texts. Of the 13 subreddits, 5 were directly associated with a mental illness: `"bipolar"`, `"adhd"`, `"anxiety"`, `"depression"`, and `"ptsd"` while the remaining were chosen from a wide range of subreddit topics and assigned the class label of `"none"`.

Previously, work such as (Shen and Rudzicz 2017) applied machine learning to classifying text by associated mental health condition.

Note that, in our work, we aim to distinguish finer-grained categories than in previous work: we are specifically interested in disintguishing posts in the Anxiety subreddit into two classes: posts from people who will and will not later post in the ADHD subreddit. That is a more difficult task than distinguishing posts in the Anxiety subreddit from posts in the ADHD subreddit, as in previous work.

| Model | (Ameer et al. 2022) | (Murarka, Radhakrishnan, and Ravichandran 2021) |
|---|---|---|
| LSTM | 76% | 72% |
| BERT | 78% | 82% |
| RoBERTa | 83% | 86% |

Table 1: Multiclass classification accuracy for Reddit posts

## Limitations

Our primary goal is to gain insight into the connection between anxiety and ADHD, as well as ADHD co-morbid with anxiety. We use a particular social media platform (Reddit), which is primarily English-speaking and whose audience is known to skew male [2]. Conclusions about symptoms drawn from Reddit therefore are likely biased by language, gender, and culture. We do not have specific demographic information about the Anxiety and ADHD subreddits, so that conclusion is itself tentative.

The classifier we train is not intended as a diagnostic tool and should not be used as such.

Classification results on text from outside of Reddit and outside of the Anxiety subreddit would likely not match what we report.

## Ethical considerations

We use public data, and as such the research is not human-subjects research, as confirmed by our IRB.

Owing to the sensitivity of the topic, we have decided not to include samples from our data in the paper.

Creating classifiers whose output is mental health conditions is fraught with the danger that such a classifier would be used without consent on users' text. Care should be taken that this does not happen. Our classifier is not diagnostically useful.

---

[2]https://www.statista.com/statistics/1255182/distribution-of-users-on-reddit-worldwide-gender/