# OpenReview forum: "LLMs Pick Up Cues of Potential Comorbid ADHD in People Reporting Anxiety when Keywords Are Not Enough"
_AAAI.org/2024/Spring_Symposium_Series/Clinical_FMs — AAAI 2024 SSS on Clinical FMs_

### Official Review · Reviewer_gphm · 2024-02-19

**Rating:** 6
**Confidence:** 4

**Review:**

Summary: This paper trained the RoBERTa model to predict whether individuals discussing anxiety in their posts will subsequently express interest in ADHD. They showed that it shows high performance (76% correct), which can give insights into their comorbidity.

Comments:
1. It is unclear what the ability of the RoBERT model to classify the groups implies. There could be many ways that drawing some clinical insights from the model performance can go wrong or be indefinite. The examples are below
- Some symptoms of anxiety are also indicative of ADHD. The RoBERT model captures the terms related to the comorbidity. However, it still seems unclear if they are just associative or if they have a causal relationship.
- ADHD patients have some features in common in their posts (not related to disorder symptoms)
- Or it could just imply selection bias as the reddit is not prospective data.
I think the implication should be more clearly stated. Also, I think modification of the experimental design could be necessary.

2. “Social media such as Reddit provides publicly available text data of anonymous
first-person experiences (Low et al. 2020).” This sentence at the end of the first paragraph in the introduction section looks abrupt. The first paragraph is mainly about the problem of misdiagnosis of ADHD and anxiety, so I think this sentence on the data source of this study should be discussed in the next paragraph.

---

### Official Review · Reviewer_sZzm · 2024-02-21
**Good research work done to evaluate efficacy of RoBERTa for a specific use case.**

**Rating:** 9
**Confidence:** 4

**Review:**

The paper provided clear understanding of problem statement, and necessary background information to understand the challenge faced by less accurate techniques for detecting co-morbid ADHD. The data used for training, however, is from a platform that is biased in representation of general population, which is also acknowledged by the authors, and also appropriately explained the limited scope of the application of the results discovered by the authors.
The quality of work is good and meets the expectation. I do not consider myself to be able to comment on the originality of the work, as I need more experience in the field to be fair in my evaluation, however, the work is fairly original in my opinion. Significance of study presented is that it provides the comparison of three different models in performing classification for the task at hand, and find a model that outperforms the other two by a significant margin.
Pro of the paper is that it has found a model that has significantly higher prediction accuracy over other models.
Con of the paper is that the data set is biased and the visualizations cannot be published to protect the patients.
However, the results of the study are significant enough to outweigh the cons. This paper deserves publication in the esteemed conference.

---

### Official Review · Reviewer_VnGH · 2024-02-22
**Online forum text based classification for a weak proxy task is better solved by RoBERTa as compared to keyword based models.**

**Rating:** 4
**Confidence:** 4

**Review:**

## Paper summary

The task of predicting whether a reddit user who first starts posting on Anxiety subreddits will later also start posting on ADHD subreddits is used as a proxy for identifying people with anxiety who might also have ADHD. Classification performance of a fine-tuned RoBERTa model is presented which is shown to be better than keyword based baselines. Some explainability experiments are promised in the future.

### Strengths
1. Misdiagnosed comorbid ADHD is an important issue.
1. Data collection and processing is sound -- the 6 month gap in user postage history between their first post in ADHD subreddits and the data gathered from anxiety subreddits is a reasonable choice.

### Weaknesses
1. The authors have acknowledged the weaknesses and assumptions in the proxy task -- subreddit posting behavior is a very weak link to whether the user actually has a high risk of having comorbid ADHD. There seems to be no method for verifying the link between this proxy task and actual comorbid ADHD.
1. The practical benefits of how the proposed study can better enable diagnosis of comorbid ADHD in the future is not discussed.

### Feedback to authors
* In order to refine the ground truth for the proxy task, a Chat-GPT or equivalent LLM can be used to query whether the user believes that they have ADHD and/or anxiety from their posts alone. This may clean up the collected data significantly.

---

### Official Review · Reviewer_LMhd · 2024-02-23
**The authors describe their goal to identify a proxy of comorbid ADHD through reddit posts in the Anxiety and ADHD subreddits. The object to find posters that initially post in the Anxiety then in the ADHD subreddit is solved much better with a finetuned RoBERTa than the baseline when using keywords.**

**Rating:** 2
**Confidence:** 4

**Review:**

Clarity: The authors clearly describe their goal of assessing two groups of reddit posters the posters in the Anxiety subreddit and that then post in the ADHD subreddit and the posters in the Anxiety that do not post in the ADHD subreddit.
Originality: the task of determining a proxy of possible comorbid ADHD is novel.
Significance: The link between the proxy of possible comorbid ADHD is not well discussed in the manuscript and therefore it is hard to say what impact this will have in the clinical world. However, it is an interesting approach to the use of foundation models within a ‘semi’-clinical setting.
Quality: the study quality appears good as the method, data and performance of the model has been clearly stated.

Major points
1.	You point towards it yourself in the data collection section of the paper where you note that the reddit posts are not a clinical diagnosis. I think you must discuss the implications on the significance of your work.
2.	Why did you choose to remove data from posters that posted in the ADHD subreddit within 6 months after their post in the Anxiety subreddit?
3.	Clarify if you download posts from the ADHD subreddit. In the data preprocessing section, you state that you don’t use the posts and then that you use the posts from the ADHD subreddit.
4.	You reference a figure 6 that is not present in the manuscript.
5.	You state that you visualize the phrases leading to “will post in ADHD” or “will not post in ADHD” for any given post but this is not presented anywhere.

Minor points
1.	In section Data Preprocessing is it correctly understood that the posters who posted anywhere else than the Anxiety or/and ADHD subreddits were removed from the dataset? This should be clearer.
2.	You mention the base rate of the test set, but you provide no detail on the distribution of the training set or if you have done any weighted sampling or indeed how you sampled the test dataset.

Pros
•	Well written and concise.
•	Interesting subject sure to spark interest even for people who are not experts in psychiatric disorders.
•	Interesting application of existing models to proxy ADHD comorbidity
•	In the appendix of the paper the limitations section does a good job of explaining the cons of the study in non-bias manner.
Cons
•	The ADHD comorbidity proxy is not well discussed.
•	The authors state that they have visualizations multiple times where none are shown.